# A validation of the Dutch version of the Awareness of Narrative Identity Questionnaire (ANIQ-NL)

**Nadischa Helena Dierdorp**[1]*, **Elien Vanderveren**[2], **David John Hallford**[3], **Dirk Hermans**[4]

**1** Clinical Psychology, KU Leuven, Leuven, Belgium, **2** School Psychology and Development in Context, KU Leuven, Leuven, Belgium, **3** School of Psychology, Deakin University, Melbourne, Victoria, Australia, **4** Centre for the Psychology of Learning and Experimental Psychopathology, KU Leuven, Leuven, Belgium

* Nadischa.Dierdorp@kuleuven.be

**Data Availability Statement:** All data files are available from the Open Science Framework database (access via https://osf.io/wn6rf/).

## Abstract

Individuals build a narrative identity through the construction of an internalised, unfolding life story based on significant autobiographical memories. The current study validated a Dutch version of the Awareness of Narrative Identity Questionnaire (ANIQ-NL), which assesses how aware individuals are of having a narrative identity as well as their perception of the global coherence within their autobiographical memories, specifically, in terms of temporal ordering, causal connections and thematic integration. The questionnaire was administered to 541 adults (65.1% female, $M_{age}$ = 34.09, $SD_{age}$ = 15.04, age range = 18–75). The results of a confirmatory factor analysis provided evidence for a four-factor structure, consisting of awareness and the three coherence subscales. The factor loadings of the items varied between .67 and .96. Moreover, the ANIQ-NL subscales showed good to excellent internal consistency, with Cronbach's alphas ranging from .86 to .96. Furthermore, higher levels of perceived autobiographical memory coherence were found to be significantly correlated to lower levels of depression, anxiety, and stress symptoms. The ANIQ-NL was determined to be a valid and reliable tool to measure narrative identity awareness and perceived narrative coherence. Future research could utilise the ANIQ-NL to further investigate the role of narrative identity in psychological well-being.

## Introduction

Individuals develop a narrative identity by integrating past personal experiences into an internalised, unfolding story of life [1–6]. In everyday life, we shape positive and negative, congruent and contradictory experiences into cohesive stories about the self and narrate these to ourselves and others. As we do this, we give meaning to experiences in our lives and try to understand ourselves as a unique individual and as a social being. By making meaningful connections in consciousness between past personal experiences, a sense of self psychologically arises [3, 7–9]. Individuals reminisce and reflect on their personal experiences to learn who they are through time. A narrative identity therefore consists of fundamental self-knowledge about who we were in the past, who we are now, and how we see ourselves in the future. This

**Funding:** The authors received no specific funding for this work.

**Competing interests:** The authors have declared that no competing interests exist.

critical form of self-knowledge allows us to have the experience of being a coherent person over time and provides us with a sense of meaning and purpose in life [3, 8–10].

Written or told autobiographical memories are often used as a methodological tool to expand our understanding of this powerful, uniquely human process of self-authorship called narrative identity. Single autobiographical memories as well as life stories consist of various characteristics [11]. Recent studies emphasise the importance of coherence, or structure, within these narratives. Coherence appears to be both an indicator and a predictor of psychological well-being. It has been found that the level of coherence within our personal narratives about life events is related to our success in extracting meaning from these events [12, 13]. Moreover, lower levels of narrative coherence have repeatedly been shown to be related to and be predictive of the development of (symptoms of) internalising psychopathology, such as depression, anxiety, and stress [14–19], as well as externalising behaviour problems in children [19–22]. In addition, the coherence within our personal narratives is related to our social well-being [23–26] and identity functioning [27]. Narrative coherence is, for example, found to be impaired in self-disorders such as schizophrenia [28] and borderline personality disorder [29, 30], and found to be negatively related to symptoms of antisocial personality disorder [31], disorders in which the sense of self is disturbed and identity is diffuse. Similarly, narrative coherence has been shown to have a buffering effect on the negative impact of psychosocial and traumatic stressors in life [19, 32–34] and can therefore be regarded a mechanism of resilience. Yet, the link between expressed narrative coherence and psychological well-being has not always been replicated and contradictions are found [18, 27, 35–37], which suggests the association may not be simply linear and straightforward.

To create a coherent life story, Habermas and Bluck [38] distinguish four subtypes of necessary narrative coherence: temporal, causal, thematic, and cultural coherence. First, the remembered events must be temporally ordered. The narrator may use temporal linguistic markers, such as conjunctions or adverbs, or may explicitly use dates of events or cross-references to historically known events or other parts of the life story. Second, the life story must contain causal links between events or chapters in life. Causal coherence also refers to the capability of the narrator to explain changes in one's values or personality as a result of events throughout life. Causal coherence is a central, essential form of coherence, because without causal links life might seem to be determined by chance and it may be harder to create or identify meaning. Third, thematic coherence is the ability to recognise similarities, or overarching themes, between different elements in the life story. Thematic coherence involves making explicit evaluative statements about what happened in life. The narrator may also compare events or describe life in terms of an evaluative trajectory. By doing this, the narrator recognises that it is important to interpret life to give it meaning. Last, cultural coherence refers to cultural norms that inform us about how to build an appropriate life story, for example, which facts and events one's life story should contain and what represents an important event in one's story about the self.

Autobiographical reasoning evolves in social interactions and is therefore fundamentally a social-cultural process [39, 40]. Each type of coherence has its own characteristic developmental trajectory and builds upon social-cognitive abilities (e.g., perspective taking, creating overarching temporal links, capacity for self-reflection) [38, 41]. While pre-schoolers and primary school children can generate a narrative that is temporally ordered, it takes until adolescence for them to create a more advanced (i.e., causal and thematic) coherent narrative, necessary for identity development. Developing coherent, causally connected and thematically interpreted, narratives is a continuous process during adulthood, as the understanding and interpretation of life events evolves along with related identity exploration [40, 41].

More recently, research has highlighted the importance of another dimension of narrative identity: *awareness* of narrative identity [1]. This is described as the individual's awareness that

their memories of past experiences are like stories that can help understand one's identity. This higher order awareness refers to the process of consciously drawing on these stories to make sense of one's life and make predictions about the future. Being aware of having life stories is found to be adaptive, irrespective of the actual coherence within these stories. Hallford and Mellor [1] observed that individuals who are more aware of having a narrative identity, more often appeal to their autobiographical memories to achieve self-continuity, think and talk about their lives more often and experience more meaning in life. Individuals who are more aware of having life stories also have a higher self-esteem and experience a greater sense of competence (i.e., have a more positive self-image and personal resources) [1]. Being aware of how past experiences, represented as stories about the self, inform one about one's identity might be fundamental to work out the kind of person one is.

To date, narrative coherence has been mainly assessed through the expression of coherence in a written or told narrative. Participants are prompted to recall a significant experience, such as a high or low point (i.e., a very positive or negative event in one's life), self-defining memory or turning point (i.e., an event that has changed one's life or the kind of person one is). To quantify and subsequently analyse the narratives and their association with a variety of psychological variables, an independent observer (transcribes and) codes the narratives for various dimensions, such as chronological and thematic cohesion. However, Hallford and Mellor [1] developed and validated the *Awareness of Narrative Identity Questionnaire (ANIQ)*, a self-report tool that assesses individuals' perception of how globally coherent one's autobiographical memories are, along with individuals' metacognitive awareness of having a narrative identity. The questionnaire consists of four dimensions: one that measures awareness and three that measure narrative coherence in terms of temporal order, causal associations and integrated themes. Cultural coherence was not included, as it is presumably interwoven with the social norms of what is understood by temporal, causal, and thematic coherence [1]. By developing the ANIQ, Hallford and Mellor provide an additional, quantitative approach to assess narrative coherence. Subsequent research could use the ANIQ to further explore how awareness of narrative identity and the perceived ability to reason with autobiographical memories are related to psychological well-being. Moreover, the ANIQ has already been used as a tool to gain insight into changes in the awareness of narrative identity in the context of psychotherapy. It has been found that individuals become more aware of having a narrative identity as a result of participating in, for example, brief reminiscence activities or a three-session group-based reminiscence program, and that this has a positive impact on their self-concept, meaning making, and psychological well-being (i.e., fewer symptoms of depression, anxiety, and stress) [42, 43]. In addition, research using the ANIQ has shown that the level of perceived causal coherence within life stories predicted later depressive symptoms, and that this could be explained by changes in self-concept [44].

Meanwhile, the ANIQ has successfully been translated and validated in Polish [45] and Turkish [46]. The researchers conducted a psychometric comparison between the original version and the translated version to confirm their equivalence. To the best of the authors' knowledge, there is no Dutch-language instrument yet that measures narrative identity awareness and the perceived ability to create coherent stories about the self. The aim of this study is therefore to adapt the Awareness of Narrative Identity Questionnaire into Dutch.

## Psychometric properties of the original version of the Awareness of Narrative Identity Questionnaire

Hallford and Mellor [1] developed and validated subscales to measure awareness of narrative identity and perceived narrative coherence. After applying an exploratory factor analysis

(EFA) on an initial item pool, Hallford and Mellor established a simple 20-item four-factor structure. In a second study, they found evidence for stability of the factor structure using a confirmatory factor analysis (CFA). Regarding reliability, the internal consistency among the items within their subscales was excellent and the one-week test-retest reliability of the subscales was found to be good. Moreover, it was found that higher scores on all three coherence subscales were able to predict higher scores on the (higher order) awareness subscale, as hypothesised. Results showed that age was slightly positively correlated with awareness and causal coherence. No gender differences were found.

Furthermore, results indicated that higher levels of awareness of narrative identity and narrative coherence, subjectively perceived in this case, are related to a more positive self-concept (see above) and lower levels of depressive and anxious symptoms [1]. In a third study, the authors found evidence that the ANIQ coherence subscales have criterion validity, as the scores on the subscales were significantly correlated with the more objective measure of narrative coherence provided through written turning-point narratives. Moreover, Janowicz et al. [47] found a significant correlation between the ANIQ awareness subscale and the level of coded coherence in written accounts of important interpersonal relationships.

In the current study, the validity and reliability of the Dutch version of the Awareness of Narrative Identity Questionnaire (ANIQ-NL) will be investigated. The original psychometric findings are expected to be reconfirmed. With respect to convergent validity, it is hypothesised that higher levels of awareness of narrative identity and, subjectively perceived, narrative coherence are negatively related to and able to predict unique variance in the levels of depression, anxiety, and stress symptoms.

## Methods

### Participants

After a process of data cleaning (removing invalid responses, i.e., demographics invalidly entered or incomplete ANIQ-NL forms), the final sample for analysis included 541 participants between the ages of 18 and 75 ($M$ = 34.09, $SD$ = 15.04), of which 65.1% were female, 34.8% were male, and 0.2% identified as otherwise. With respect to highest education level obtained, 2.2% obtained a doctorate, 30.9% a university master's degree, 15% a university bachelor's degree, 22.9% a non-university bachelor's degree, 24.8% had completed higher secondary education, 3.9% had completed lower secondary education, and 0.4% had completed primary education. Furthermore, 37.5% of the sample reported currently studying, 35.7% working full-time, 12.8% working part-time, 7.2% were unemployed, and 6.8% were retired. Regarding romantic relationships, 39% of the participants were single, 18.7% had a romantic partner but did not live together, 17.7% had a romantic partner and were cohabiting, and 24.6% were married. Males and females did not differ in demographic characteristics, including their age, $t(537)$ = 1.42, $p$ = .157, $d$ = 0.12, education level, $t(356.33)$ = -0.79, $p$ = .430, $d$ = -0.07, current work status, $t(538)$ = -0.01, $p$ = .991, $d$ = -0.00, and marital status, $t(538)$ = 0.55, $p$ = .581, $d$ = 0.05.

### Materials

**Awareness of Narrative Identity Questionnaire (ANIQ).** The ANIQ consists of 20 items and has four subscales: Awareness, Temporal Coherence, Causal Coherence, and Thematic Coherence. Awareness refers to the extent to which one is aware of having a narrative identity, for example, *"When I think over my life, I can observe how there is a story that tells me who I am"*. Temporal Coherence is the ability to perceive the order in which life events took place (e.g., *"I can put the events of my life in order of when they occurred"*). Causal Coherence refers

to the understanding of how experiences in life are interrelated and causally connected to one another (e.g., *"Things that have happened over the course of my life are meaningfully tied together"*). Thematic Coherence is the ability to identify overarching themes in one's personal memories that relate to the kind of person one is (e.g., *"When I think or talk about experiences in my past I can see themes about the kind of person I am"*). Each subscale consists of five self-report items. Participants respond to the items on an 11-point Likert scale, ranging from *'completely disagree'* (0) to *'completely agree'* (10). The scores of all items per subscale are added together to calculate the scale score. All scale scores are then added together to arrive at a total score. Hallford and Mellor found that the ANIQ has good validity and reliability [1]. In all three studies conducted, both internal validity ($\alpha$ = .86–96) and test-retest reliability ($r$ = .72-.79) were found to be high. The Polish [45] and Turkish [46] translations of the ANIQ also demonstrated favourable psychometric properties. The Polish version displayed Cronbach's alphas ranging from .87 to .94 and test-retest coefficients between .58 and .91. As for the Turkish version, Cronbach's alphas varied from .84 to .94, and the test-retest coefficients ranged from .77 and .95.

**Depression Anxiety Stress Scales (DASS-21).** To assess symptoms of psychological distress, the short-form of the Dutch version of the *Depression, Anxiety and Stress Scales* (DASS-21) [48, 49] was used. The questionnaire contains three subscales of seven items each, which measure core symptoms of depression, anxiety, and stress. Participants indicate to what extent the items applied to them over the past week using a 4-point Likert scale ranging from *'not at all or never'* (0) to *'surely or most of the time'* (3). The DASS-21 has a total of 21 items. The questionnaire possesses good validity and internal reliability [48–50]. Studies have previously reported a Cronbach's alpha of .91 for Depression, .86 for Anxiety, and .85 for Stress [48]. Internal reliability was also found to be good in the current study, with Cronbach's alphas of .89 for Depression, .82 for Anxiety, and .86 for Stress.

**Translation process.** The translation process of the original English version of the ANIQ into Dutch (ANIQ-NL) involved multiple steps. Initially, three independent translators, all native Dutch speakers, conducted a forward translation of the questionnaire from English to Dutch. One of the translators had a background in healthcare and was familiar with the terminology and construct matter in Dutch, while the other two translators were knowledgeable about informal expressions, health-care specific language, idiomatic phrases, and commonly used emotional terms in Dutch. Subsequently, the three translated versions of the instrument were compared with the support of a fourth independent translator who is an expert in the field of autobiographical memory coherence. Any ambiguities or discrepancies were discussed and resolved through a committee approach. Once the four translators reached consensus, the preliminary Dutch version was blind back-translated by experts from the Interfaculty Institute for Living Languages (ILT) at KU Leuven. Finally, a multidisciplinary committee, consisting of the four translators involved in the forward translation and the developer of the original questionnaire, David John Hallford, evaluated the translation for its accuracy. The back-translation and the original version of the questionnaire were compared to assess semantic equivalence. It turned out that the items in the Dutch version retained the intended meaning of the English items (see S1 Appendix for the final version of the ANIQ-NL).

## Procedure

The data collection took place at two time-points: the first part was collected in February 2021 ($n$ = 415) and the second in February 2022 ($n$ = 126). In both waves of data collection, participants were recruited through flyers in public places and via online social media platforms. To participate in the study, participants had to be at least 18 years old as assessed by self-report.

No other inclusion or exclusion criteria were used. Participation in the study was on a voluntary basis. As compensation, for every 50 participants one 50 Euro gift voucher was randomly awarded to a participant. The data was collected online using Qualtrics. Here, participants first received information about the processing of personal data, in accordance with the General Data Protection Regulation (GDPR). After this, they were informed about the purpose of the study and the conditions of participation and were asked to provide their informed consent (by selecting 'yes'). To be eligible for the voucher, participants were asked to enter their email address so the researchers could issue the vouchers. Thereafter, demographics were requested and, subsequently, all participants completed the ANIQ-NL and DASS-21. After data collection, the email addresses have directly been removed from the original data and stored separately so that the authors did not have access to information that could possibly identify individual participants during data analysis. This study was approved by the Social and Societal Ethics Committee of the KU Leuven (approval numbers G-2020-2745 and G-2021-4035-R3) and (only the second part of the data collection) was preregistered on AsPredicted.org (https://aspredicted.org/xg38k.pdf).

## Data analysis

To verify construct validity of the ANIQ-NL, CFAs were performed to test and contrast models. Structural equation models (SEMs) with maximum likelihood estimators were applied. Several indices were used to assess the fit between the proposed four-factor model and the current data, including the relative Chi-square goodness-of-fit (GOF) statistic ($\chi^2$/$df$ ratio) and associated $p$ value, the root mean square error of approximation (RMSEA), the standardised root mean square residual (SRMR), and the comparative fit index (CFI). The following benchmarks were used for the interpretation of good model fit on these indices: a value less than 5 – and ideally less than 2 [51]–for the $\chi^2$/$df$ ratio [52], values less than or equal to .08 for the RMSEA and SRMR, and a value equal to or greater than .95 for the CFI [53]. Cronbach's alphas and composite reliability scores (CR) were used to assess the internal consistency of the subscales. In addition to Cronbach's alpha, CR is a measure that takes into account the factor loadings of the items and allows for correlated errors [54–56]. The following rules of thumb are used to interpret Cronbach's alphas: < .50 is unacceptable, ≥ .50 is poor, > .60 is questionable, > .70 is acceptable, > .80 is good, and > .90 is excellent [57]. Similarly, CR values above 0.70 are generally considered acceptable [54]. To evaluate the convergent validity of the ANIQ-NL subscales, average variance extracted (AVE) scores were computed. AVE represents the proportion of variance captured by the latent factor in relation to the measurement error. AVE scores of 0.50 or higher are typically considered acceptable [54]. Pearson correlations were used to assess associations between the ANIQ-NL subscales, as well as associations between the ANIQ-NL subscales and symptoms of psychological distress and age. Independent $t$-tests (two-tailed) were used to assess gender differences on the ANIQ-NL subscales. Multiple regression analyses were conducted to assess whether the three Coherence subscales predicted unique variance in the level of Awareness and whether the ANIQ-NL subscales predicted unique variance in the symptoms of psychological distress. An alpha-level of .05 was set for all analyses. Prior to conducting data analysis, it was confirmed that the scores on the ANIQ-NL subscales followed a normal distribution, which is a prerequisite for conducting SEM and linear regression analyses. Considering the exploratory nature of the study, outliers were retained in the dataset to ensure a rigorous examination of the psychometric properties of the ANIQ-NL. It is worth mentioning that some participants (n = 6) provided linear responses, i.e., all ANIQ-NL items were given the same answer, resulting in exceptionally high scores on the ANIQ-NL subscales. Therefore, all analyses were reconducted after removal of

linear responses. No meaningful changes in the results were observed. The data analyses were performed with IBM SPSS Statistics 28.0.1.1 and JASP 0.16.1.

## Results

### Construct validity

A CFA was performed to determine the fit of the proposed four-factor model. The four-factor model, with a $\chi^2/df$ ratio of 3.97, a CFI of .95, a RMSEA of .07, and a SRMR of .05, fitted the data well. The $p$-value of the Chi-square goodness-of-fit statistic was significant, but this was likely an artefact of using a large sample [58, 59].

To assess whether more parsimonious factor structures would fit the data as well, additional CFAs were conducted. Both a two-factor model consisting of the factor Awareness (item 1 to 5) and the factor Coherence, on which all coherence items are loaded (item 6 to 20), and a one-factor model (item 1 to 20) were evaluated. Table 1 summarises the fit indices for all tested models. As can be seen in Table 1, the fit indices of the two- and one-factor model showed an unacceptable model fit. The results confirmed that the four-factor model fits the data best.

Within the four-factor model, all items loaded strongly onto their respective factors. One exception was Item 3, which loaded only moderately onto its factor. Factor loadings ranged from .67 to .80 for the Awareness subscale, from .85 to .96 for the Temporal Coherence subscale, from .74 to .87 for the Causal Coherence subscale, and from .71 to .90 for the Thematic Coherence subscale. All factor loadings were found to be highly statistically significant ($p <$ .001). Table 2 provides an overview of the factor loadings, along with the means and standard deviations of the items, subscales, and total score.

Furthermore, as shown in Table 2, internal consistency among the items within the subscales was found to be good to excellent, with Cronbach's alphas of .86 for Awareness, .96 for Temporal Coherence, .90 for Causal Coherence, .92 for Thematic Coherence, and .94 for the ANIQ-NL total score. The CR scores provided additional support for a high level of internal consistency, with values ranging from .81 to .99. This suggests that the items within each subscale consistently and reliably measured the underlying construct. Furthermore, the AVE scores for all subscales exceeded the recommended threshold of 0.50, indicating that the items accounted for a substantial proportion of the variance within their respective constructs. Moreover, all subscales were significantly correlated with each other and the total score. The strength of the correlation with the Awareness subscale significantly increased from Temporal to Causal Coherence, $Z(538) = 4.36$, $p < .001$, and from Causal to Thematic Coherence, $Z(538) = 6.32$, $p < .001$, as they each represent increasingly higher-order forms of autobiographical reasoning. Fig 1 provides an overview of the four-factor model along with the correlations among the subscales.

**Table 1. Fit indices for CFAs on ANIQ-NL items.**

| Model | $\chi^2$ | $df$ | $\chi^2/df$ | $p$ | RMSEA | RMSEA [90% CI] | SRMR | CFI |
|---|---|---|---|---|---|---|---|---|
| One-factor model (current study) | 4256.20 | 170 | 25.04 | < .001 | .21 | [.21, .22] | .14 | .55 |
| Two-factor model (current study) | 3772.19 | 169 | 22.32 | < .001 | .20 | [.19, .20] | .13 | .61 |
| Four-factor model (current study) | 651.71 | 164 | 3.97 | < .001 | .07 | [.07, .08] | .05 | .95 |
| Four-factor model (Hallford & Mellor, 2017) | 400.7 | 163 | 2.46 | < .001 | .07 | [.06, .08] | .04 | .95 |

$\chi^2$ = Chi-square value; $df$ = degrees of freedom; $\chi^2/df$ = relative Chi-square statistic; RMSEA = root mean square error of approximation; CI = confidence interval; SRMR = standardised root mean square residual; CFI = comparative fit index.

**Table 2. Factor loadings and descriptive statistics for ANIQ-NL items and subscales.**

| Item | Awareness | Temporal Coherence | Causal Coherence | Thematic Coherence | Total | Item scores, $M$ ($SD$) |
|---|---|---|---|---|---|---|
| 1 | .78 | - | - | - | - | 7.0 (2.0) |
| 2 | .80 | - | - | - | - | 6.5 (2.1) |
| 3 | .67 | - | - | - | - | 7.4 (2.0) |
| 4 | .71 | - | - | - | - | 6.6 (2.0) |
| 5 | .75 | - | - | - | - | 6.3 (2.1) |
| 6 | - | .89 | - | - | - | 7.2 (2.3) |
| 7 | - | .93 | - | - | - | 7.1 (2.3) |
| 8 | - | .91 | - | - | - | 7.1 (2.1) |
| 9 | - | .96 | - | - | - | 7.1 (2.2) |
| 10 | - | .85 | - | - | - | 6.6 (2.2) |
| 11 | - | - | .74 | - | - | 6.6 (1.9) |
| 12 | - | - | .82 | - | - | 6.9 (1.9) |
| 13 | - | - | .81 | - | - | 6.5 (1.9) |
| 14 | - | - | .87 | - | - | 6.6 (1.9) |
| 15 | - | - | .81 | - | - | 7.0 (1.8) |
| 16 | - | - | - | .86 | - | 6.8 (2.0) |
| 17 | - | - | - | .90 | - | 6.8 (1.9) |
| 18 | - | - | - | .90 | - | 6.9 (1.9) |
| 19 | - | - | - | .71 | - | 7.1 (1.9) |
| 20 | - | - | - | .85 | - | 7.1 (1.9) |
| Subscale totals, $M$ ($SD$) | 33.8 (8.1) | 35.1 (10.2) | 33.6 (8.0) | 34.7 (8.5) | 137.2 (27.5) | |
| AVE | .50 | .81 | .68 | .73 | .65 | |
| CR | .81 | .99 | .99 | .99 | .96 | |
| Cronbach's α (current study) | .86 | .96 | .90 | .92 | .94 | |
| Cronbach's α (Hallford & Mellor, 2017) | .91 | .96 | .90 | .93 | - | |

All item loadings are significant at the $p < .001$ level. Item range is 1 to 10, subscale range is 5 to 50, and total range is 20 to 200.

A multiple regression analysis was performed to investigate whether the different types of perceived coherence in one's life stories uniquely contribute to a greater awareness of narrative identity. The multiple regression analysis with the Awareness subscale as outcome variable and the three Coherence subscales entered simultaneously into the model as predictor variables was significant, $F(3, 537) = 141.77$, $p < .001$, and explained 44.2% (adjusted $R^2 = 43.9\%$) of the variance in the Awareness subscale. In the model, Thematic Coherence was found to be an independent significant predictor of greater awareness of having a narrative identity, standardised $\beta = .61$, $p < .001$, 95% CI = .50, .67, whereas this was not the case for Temporal Coherence, standardised $\beta = .03$, $p = .430$, 95% CI = -.04, .08, or Causal Coherence, standardised $\beta = .06$, $p = .200$, 95% CI = -.03, .15.

Finally, it was explored whether gender and age are related to differences in the level of awareness of narrative identity and perceived narrative coherence. Results showed that gender has a small effect on the level of Awareness, Causal Coherence, and Thematic Coherence, $t(538) = 3.10$, $p = .002$, $t(538) = 2.30$, $p = .022$, $t(538) = 3.32$, $p < .001$, with women scoring higher than men. No gender differences were observed for Temporal Coherence, $t(420.27) = 0.62$, $p = .533$. Moreover, age correlated with the level of Causal Coherence, $r = .18$, $p < .001$. As the age of participants increased, they reported perceiving their life stories as more causally coherent. Table 3 shows the findings regarding gender and age.

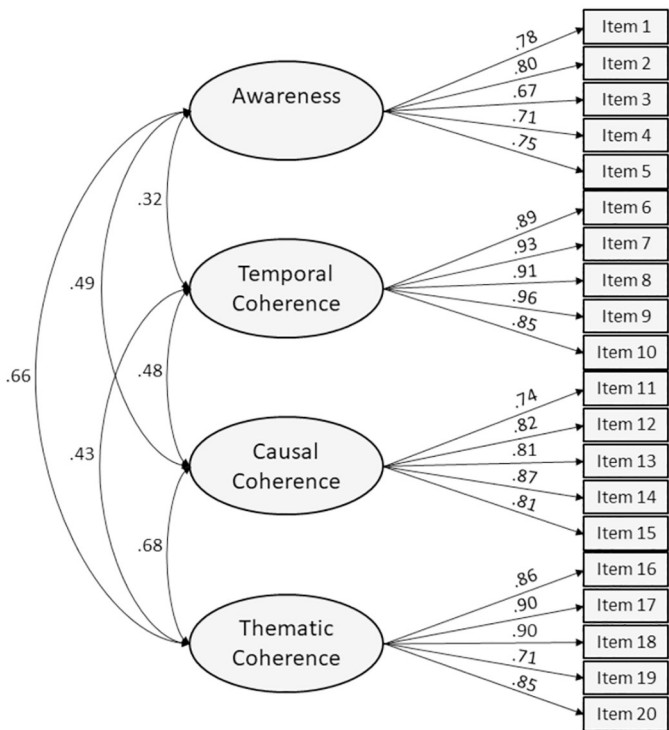

**Fig 1. Four-factor model of the ANIQ-NL with standardised factor loadings.** All factor loadings and correlations are significant at the $p < .001$ level.

## Convergent validity

Regarding convergent validity, the negative relationship between the ANIQ-NL and levels of depression, anxiety, and stress symptoms was examined. As shown in Table 4, lower scores on the Temporal, Causal and Thematic Coherence subscales were found to be significantly correlated with higher levels of depression, anxiety, and stress symptoms, except for Thematic Coherence and symptoms of anxiety and stress. The level of Awareness was not correlated with symptoms of psychological distress.

To assess whether scores on the ANIQ (subscales) predict unique variance in the level of psychological distress, a multiple regression analysis was performed. The multiple regression

**Table 3. Gender differences for mean scores on ANIQ-NL subscales and correlations with age.**

|  | Men, M (SD) | Women, M (SD) | Cohen's d | Age |
|---|---|---|---|---|
| Awareness | 32.4 (7.9) | 34.6 (8.0) | 0.28** | .03 |
| Temporal Coherence | 34.8 (9.4) | 35.3 (10.5) | 0.05 | .06 |
| Causal Coherence | 32.6 (7.6) | 34.2 (8.1) | 0.21* | .18*** |
| Thematic Coherence | 33.1 (8.0) | 35.6 (8.6) | 0.30*** | .05 |
| ANIT-NL Total | 132.8 (24.5) | 139.7 (28.5) | 0.26** | .10* |

Subscale range is 5 to 50, and total range is 20 to 200.

*$p < .05$

**$p < .01$

***$p < .001$

**Table 4. Correlations between ANIQ-NL subscales and DASS-21 subscales.**

|  | Awareness | Temporal Coherence | Causal Coherence | Thematic Coherence | ANIQ-NL Total | Subscale totals, $M$ ($SD$) |
|---|---|---|---|---|---|---|
| Depression | -.07 | -.18*** | -.23*** | -.16*** | -.20*** | 6.0 (4.7) |
| Anxiety | .00 | -.15*** | -.15*** | -.08 | -.13** | 4.1 (3.7) |
| Stress | .01 | -.13** | -.12** | -.04 | -.09* | 7.7 (4.2) |
| DASS-21 Total | -.02 | -.18*** | -.19*** | -.11* | -.16*** | 17.9 (11.0) |

DASS-21 = Depression, Anxiety and Stress Scales, 21 item version. DASS-21 subscale range is 0 to 21, and total range is 0 to 63.

*$p < .05$

**$p < .01$

***$p < .001$

analysis with the DASS-21 total score as outcome variable and the Awareness and three Coherence subscales entered simultaneously into the model as predictor variables was significant, $F(4, 533) = 7.70$, $p < .001$, and explained 5.5% (adjusted $R^2 = 4.8\%$) of the variance in the DASS-21 total score. Temporal Coherence, standardised $\beta = -.12$, $p = .014$, 95% CI = -.23, -.03, and Causal Coherence, standardised $\beta = -.19$, $p = .002$, 95% CI = -.42, -.10, were found to be independent significant predictors of (experiencing fewer) symptoms of depression, anxiety, and stress. In contrast, Awareness, standardised $\beta = .11$, $p = .063$, 95% CI = -.01, .29, and Thematic Coherence, standardised $\beta = .00$, $p = .998$, 95% CI = -.17, .17, were not. Inspection of the standardised $\beta$ of Awareness, which turned reversed in direction relative to its zero-order correlation, suggested that there may have been statistical suppression [60, 61].

## Discussion

To date, many studies in the field of narrative identity have examined narrative coherence in terms of the ability to express coherent narratives about significant life events. In contrast, the Awareness of Narrative Identity Questionnaire (ANIQ) is a quantitative self-report tool that assesses the awareness of narrative identity and the ability to perceive stories about the self as globally coherent. The current study aimed to translate and validate a Dutch version of the ANIQ (ANIQ-NL).

The results showed that the hypothesised four-factor model fitted the data best. Within this four-factor model, the items generally loaded strongly onto their respective factors. In addition, the ANIQ subscales were found to have good to excellent internal consistency. Across all subscales, the items were able to explain a favourable amount of variance within their respective subscales. Moreover, all subscales were found to be significantly correlated with each other, although the strength of the correlations among the subscales differed, i.e., the strength of the correlation with the Awareness subscale significantly increased from Temporal, to Causal, to Thematic Coherence. This may be because they are considered to be progressively higher-order levels of autobiographical reasoning [38] that, respectively, would be more strongly associated with the notion of having life stories. Furthermore, the results of a multiple regression analysis provided evidence that higher levels of perceived coherence in stories about the self contribute to being more aware of having a narrative identity. Moreover, only the level of Thematic Coherence was able to independently predict unique variance in the degree of awareness of having a narrative identity. This is somewhat inconsistent with previous findings that all three coherence subscales contributed to unique variance to the awareness subscale [1]. However, it may reflect that the ability to perceive thematic coherence necessitates the perception of temporal and causal coherence and is conceptually most closely related to

awareness of narrative identity. Altogether, these results confirm that the ANIQ-NL subscales are representative of different forms of autobiographical reasoning that are distinctive from each other, but also closely related.

Moreover, it was explored whether gender and age have an influence on the level of awareness of narrative identity and perceived narrative coherence. Regarding gender, it was found that women scored significantly higher than men in terms of awareness of narrative identity and perceived causal and thematic coherence. This was not observed by Hallford and Mellor [1]. Nevertheless, other researchers assessing the degree of (expressed) narrative coherence have found gender differences before [27, 35]. Moreover, age correlated significantly with the level of perceived causal coherence. The older the participant, the more causally coherent they perceived themselves. This is consistent with theory about the development of narrative coherence that states that causal and thematic coherence continue to develop throughout adulthood [40, 41]. Nevertheless, no significant correlation between age and thematic coherence was observed.

Regarding convergent validity, a negative relationship between the ANIQ-NL and levels of depression, anxiety and stress symptoms was hypothesised. Lower scores on all three perceived coherence subscales were found to be significantly correlated with higher levels of depression, anxiety, and stress symptoms, except for perceived Thematic Coherence and symptoms of anxiety and stress. The level of awareness was observed to be uncorrelated with symptoms of psychological distress. Hallford et al. [1, 44] also found that awareness of narrative identity was not related to depressive and anxious symptoms. However, awareness of narrative identity, or having the notion that one's past experiences are memorised as stories that help understand one's identity, is found to be related to meaning in life [1, 44], that may in turn influence psychological well-being outcomes, such as social well-being or life satisfaction.

Using a multiple regression analysis, it was found that the ANIQ-NL subscales together were able to predict unique variance in the level of psychological distress. Both Temporal and Causal Coherence independently predicted unique variance in the levels of psychological distress. In contrast, Awareness and Thematic Coherence did not. Although, empirically, research has shown that the coherence of narratives is generally related to and predictive of an individual's psychological well-being, several studies showed that the more structural dimensions, such as causal coherence, may be most crucial for associations with psychological well-being [11, 15, 41, 62, 63]. As previously mentioned, Hallford et al. [44] also found evidence for the importance of (perceived) causal coherence. It may be reasoned that thematic coherence does not predict unique variance in the level of depression, anxiety, and stress symptoms, when temporal and causal coherence are concurrently assessed [15, 44]. Moreover, causal coherence might involve the integration of temporal knowledge, and therefore their variance may be shared to some extent. In this context, the current results suggest that while identifying overarching themes and meaning-making aspects may be important in psychological well-being, being able to temporally order and causally link life events along with making connections between life events and one's self-concept are especially important in preventing (or reducing) symptoms of depression, anxiety, and stress.

A few limitations need to be considered. First, the cross-sectional nature of this study does not allow to identify the direction of the relationship between the ANIQ Coherence subscales and the level of internalising symptoms, nor does it allow to draw conclusions regarding the causality of this relationship. Future studies with a longitudinal design would be useful, especially in the period from early adolescence to adulthood, which is known as a key period for identity development and at the same time as a period with increased vulnerability for the onset of (internalizing) psychopathology [64, 65]. Also, experimental and interventional studies on the role of awareness of narrative identity and perceived narrative coherence for

meaning making and psychological well-being are valuable [42, 43]. For example, research by Hallford et al. [43] showed that awareness of narrative identity can be regarded a mechanism of change in positive-focused Cognitive-Reminiscence Therapy (CRT), increasing resilience in young adults. Second, it is important to acknowledge certain limitations in the translation procedure. While a rigorous translation procedure was followed, it should be noted that some steps were omitted in the process [66, 67]. For example, other national experts in the field were not involved for an expert panel discussion to gather additional feedback. Furthermore, no pilot study was conducted to assess the clarity of the instructions and items in the final version of the questionnaire. Moreover, the use of a bilingual sample could have provided further evidence for conceptual, content, and construct equivalence [66, 67]. Lastly, a limitation is that due to an oversight in the digitalisation of the questionnaire, a 10-point Likert scale (ranging from 1 to 10) instead of an 11-point Likert scale (ranging from 0 to 10) was applied. However, the item mean scores and standard deviations were comparable to those observed by Hallford and Mellor [1]. In future use of the questionnaire, it is recommended to apply the originally intended 11-point Likert scale.

Altogether, the ANIQ-NL demonstrated predominantly the same psychometric qualities compared to the original version. The ANIQ-NL can therefore be considered a valid and reliable measure to assess the awareness of having life stories and perception of autobiographical memories as globally coherent. The development of this measure extends on existing measures by providing a quantitative, self-report tool that assesses the individual's subjective notion of these constructs. As previously mentioned, Hallford and Mellor [1] found support for criterion validity by establishing correlations between the scores on the ANIQ-Coherence subscales and the coherence scores derived from written turning-point narratives ($r = .23$-$.41$). Although there is some degree of correlation between these measures, a perfect overlap is not observed. This indicates that the more objective measure of narrative coherence, involving narratives coded by an independent observer, and the more subjective measure, involving the self-report questionnaire, may not necessarily align. However, this does not necessarily suggest an inherent limitation of the questionnaire itself. Researchers utilising the ANIQ may prioritise the subjective appraisal of coherence within life stories over the actual ability to demonstrate coherence. The ability to subjectively appraise coherence within our life stories may represent another aspect of our autobiographical reasoning, other than our ability to actually demonstrate coherence in narratives. This perspective is supported by the notion that higher-order awareness of life stories is likely influenced by a range of metacognitive abilities, self-awareness, and personal agency [1]. In future studies, a combination of both measures can contribute to a more comprehensive understanding of autobiographical reasoning.

Follow-up research may utilise the ANIQ-NL to further clarify the role of narrative identity in (identity) development, resilience, and psychopathology. It could be further investigated whether awareness of narrative identity (and perceived narrative coherence) is indirectly related to internalising symptoms through mediating variables, such as social resources. Furthermore, the ANIQ-NL can be used in clinical samples to investigate effects of (psycho) pathology on the awareness of narrative identity and perceived coherence within autobiographical memories. In addition, the ANIQ-NL could be used to assess change in the awareness of narrative identity and perceived narrative coherence as a result of psychotherapy.

To conclude, the Dutch version of the Awareness of Narrative Identity Questionnaire was found to be a valid and reliable measure for assessing individuals' awareness of narrative identity and self-perceived coherence within their autobiographical memories. The questionnaire could be utilised in various fields of inquiry and may help further unravel the construction of a narrative identity and its relation to psychological well-being.

## Supporting information

**S1 Appendix. ANIQ-NL.** Dutch version of the Awareness of Narrative Identity Questionnaire.
(DOCX)

## Acknowledgments

The authors would like to thank Michelle Jacobs for her support in translating and distributing the questionnaire.

## Author Contributions

**Conceptualization:** Nadischa Helena Dierdorp, Elien Vanderveren, David John Hallford, Dirk Hermans.

**Data curation:** Nadischa Helena Dierdorp.

**Formal analysis:** Nadischa Helena Dierdorp.

**Methodology:** Nadischa Helena Dierdorp, Elien Vanderveren, David John Hallford, Dirk Hermans.

**Supervision:** Elien Vanderveren, David John Hallford, Dirk Hermans.

**Writing – original draft:** Nadischa Helena Dierdorp.

**Writing – review & editing:** Nadischa Helena Dierdorp, Elien Vanderveren, David John Hallford, Dirk Hermans.

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
