## [Decision Letter · Decision Letter 0]

26 May 2023

PONE-D-23-09910A validation of the Dutch version of the Awareness of Narrative Identity Questionnaire (ANIQ-NL)PLOS ONE

Dear Dr. Dierdorp,

Thank you for submitting your manuscript to PLOS ONE. After careful consideration, we feel that it has merit but does not fully meet PLOS ONE’s publication criteria as it currently stands. Therefore, we invite you to submit a revised version of the manuscript that addresses the points raised during the review process.

We look forward to receiving your revised manuscript.

Kind regards,

Yuh-Yuh Li, Ph.D.

Academic Editor

PLOS ONE

Reviewers' comments:

Reviewer's Responses to Questions

**Comments to the Author**

1. Is the manuscript technically sound, and do the data support the conclusions?

Reviewer #1: Yes

Reviewer #2: Partly

2. Has the statistical analysis been performed appropriately and rigorously? 

Reviewer #1: Yes

Reviewer #2: No

3. Have the authors made all data underlying the findings in their manuscript fully available?

Reviewer #1: Yes

Reviewer #2: Yes

4. Is the manuscript presented in an intelligible fashion and written in standard English?

Reviewer #1: Yes

Reviewer #2: Yes

5. Review Comments to the Author

Reviewer #1: A Review of PONE-D-23-09910 "A validation of the Dutch version of the Awareness of Narrative Identity Questionnaire (ANIQ-NL)."

I recommend minor revision on the manuscript. I only have 2 minor suggestions that the authors may consider including in their revision.

In the review section and/or in the description of the ANIQ original measure, it would be informative to include reliability values and validity assessment found in the other versions. In “Hallford and Mellor found that the ANIQ has good validity and reliability [1]. The Polish [45] and Turkish [46] translations of the ANIQ have also demonstrated good psychometric properties”, specific reliability values and validity assessment details can be included there.

Something the authors might consider addressing in the discussion, was there any previous mixed-method research that investigating autobiographical life narratives together with using this developed quantitative measure? The authors may address further about the different natures of information derived from these two approaches.

Reviewer #2: Review comments on PONE-D-23-09910

Overall comments

The study design, data analysis, and presentation of the findings are scientifically rigorous. The outcome of the research is justifiable and it can benefit practitioners/researchers in the future. The four-factor model of Awareness of Narrative Identity Questionnaire-NL seems best fit as per the CFA parameters. The statistical procedures and data analysis process was adequately followed to confirm the validation of the tool.

Shortcomings: areas to be improved

- The translation of the tool and validation process was not adequately followed. From the narrative of the steps followed, a few steps such as forward translation from independent experts, consultations with experts in receiving feedback (Panel discussion with national experts on the subject), and pilot testing before the final data collection were not followed. Check more detailed procedures guided by Sousa and Rojjanasrirat (2011), and Van Ommeren et al. (1999). Such limitations are better highlighted in the limitation section so that future researchers and clinicians become aware of such caveats.

- Were there any missing data? How about the outliers?

– It is worth mentioning the data cleaning process and exclusion criteria of any data before analysis.

- Normality tests of the data are important before applying SEM. Authors are requested to describe the assurance of normality (normal distribution) of the presented data.

- Page 21: insert Fig 1.

- As authors have used CFA rather than EFA, it is advisable to present composite reliability (i.e. construct reliability) scores with average variance extracted from each sub-scale (Wang, French, & Clay, 2015). It is also recommended to add convergent validity scores as CFA is done for validation of this tool.

I thank you for this opportunity to review this interesting paper. I wish the authors may find my comments useful to improve the quality of this paper.

References:

Van Ommeren, M., Sharma, B., Thapa, S., Makaju, R., Prasain, D., Bhattarai, R., & de Jong, J. (1999). Preparing instruments for transcultural research: Use of the translation monitoring form with Nepali-speaking Bhutanese refugees. Transcultural Psychiatry, 36(3), 285-301. https://doi.org/10.1177/136346159903600304

Sousa, V. D., & Rojjanasrirat, W. (2011). Translation, adaptation and validation of instruments or scales for use in cross-cultural health care research: A clear and user-friendly guideline. Journal of Evaluation in Clinical Practice, 17(2), 268–274. https://doi.org/10.1111/j.1365-2753.2010.01434.x

Wang, X., French, B. F., & Clay, P. F. (2015). Convergent and discriminant validity with formative measurement: A mediator perspective. Journal of Modern Applied Statistical Methods, 14(1), 83-106. https://researchportal.murdoch.edu.au/esploro/outputs/journalArticle/Convergent-and-discriminant-validity-with-formative/991005544679607891

6. PLOS authors have the option to publish the peer review history of their article (what does this mean?). If published, this will include your full peer review and any attached files.

Reviewer #1: **Yes: **Nipat Pichayayothin

Reviewer #2: **Yes: **Yubaraj Adhikari, PhD

---

## [Author Response · Author response to Decision Letter 0]

14 Jun 2023

My responses to the reviewers are provided via my rebuttal letter (''Responses to Reviewer'').

---

## [Editor Report · Decision Letter 1]

15 Jun 2023

A validation of the Dutch version of the Awareness of Narrative Identity Questionnaire (ANIQ-NL)

PONE-D-23-09910R1

Dear Dr. Dierdorp,

We’re pleased to inform you that your manuscript has been judged scientifically suitable for publication and will be formally accepted for publication once it meets all outstanding technical requirements.

Kind regards,

Yuh-Yuh Li, Ph.D.

Academic Editor

PLOS ONE
---

## [Editor Report · Acceptance letter]

20 Jun 2023

PONE-D-23-09910R1 

A validation of the Dutch version of the Awareness of Narrative Identity Questionnaire (ANIQ-NL) 

Dear Dr. Dierdorp:

I'm pleased to inform you that your manuscript has been deemed suitable for publication in PLOS ONE. Congratulations! Your manuscript is now with our production department. 

Kind regards, 

on behalf of

Dr. Yuh-Yuh Li 

Academic Editor

PLOS ONE